# Effects of Polyethylene and Polystyrene Microplastics on Oat (*Avena sativa* L.) Growth and Physiological Characteristics

**DOI:** 10.3390/plants15010056

**Published:** 2025-12-24

**Authors:** Zhibo Yang, Lingping Zhao, Shitu Tan, Pei Mao, Qunying Wang, Wenfeng Ma

**Affiliations:** College of Animal Science and Technology, Henan University of Science and Technology, Luolong District, Kaiyuan Avenue, No. 263, Luoyang 471023, China; yzb20000102@163.com (Z.Y.); t_s_t@sina.com (S.T.); maop17@163.com (P.M.); 15081751970@163.com (Q.W.); a113boy@163.com (W.M.)

**Keywords:** polyethylene, polystyrene, forage, growth parameters, antioxidant enzyme activity, tolerance

## Abstract

Despite increasing environmental concerns, there are few studies on the potential effects of polyethylene and polystyrene microplastics on feed crops. The effects of polyethylene (PE) and polystyrene (PS) microplastics with a diameter of 2 μm at different concentrations (0.1%, 0.5%, 1%, and 5%) (*w*/*w*) on the growth and development of oats were analyzed in a pot experiment, with no microplastics added as the Control (Ctrl) group. The results showed that PS microplastics exhibited a spherical morphology, whereas PE microplastics displayed an irregular morphology. PE microplastics had an inhibitory effect on oat growth, chlorophyll content, photosynthetic parameters and antioxidant enzyme activity, and this effect was concentration-dependent; specifically, the inhibitory intensity increased progressively as the concentration of PE microplastics rose. In contrast, treatments involving varying concentrations of PS microplastics elicited distinct effects on the physiological and biochemical parameters of oats. The 0.1% PS microplastics treatment significantly enhanced the net photosynthetic rate of oat leaves (by 14.0%), while the 5% PS microplastics treatment significantly reduced the seedling height (by 31.1%), the total chlorophyll content (by 34.6%), the transpiration rate (by 35.7%), the stomatal conductance (by 71.1%), and the intercellular CO_2_ concentration (by 43.1%). Furthermore, a significant decrease in antioxidant enzyme activity was observed in oats after the 5% PE microplastics treatment. The activities of peroxidase (POD), catalase (CAT) and superoxide dismutase (SOD) decreased by 17.1%, 89.2% and 5.6%, respectively. At the same concentration (5%), PE microplastics exhibited a more pronounced inhibitory effect on oats compared to PS microplastics. In summary, this study demonstrates that microplastics impair photosynthesis and antioxidant capacity in oats, thereby inhibiting their normal growth and development. These findings provide a theoretical foundation and supporting data for further research into the toxicity of microplastics to oats.

## 1. Introduction

Microplastics refer to tiny plastic particles (<5 mm) found in the environment [1]. From 1950 to 2015, the world generated nearly 6.3 billion tons of plastic waste, of which only 9% was recycled, 12% was incinerated, and the remaining 79% was disposed of in landfills [2,3]. Larger plastic debris (>5 mm) is gradually fragmented by the combined effects of physical weathering, chemical oxidation, and biodegradation, ultimately forming microplastics with particle sizes < 5 mm [4]. With the widespread use of plastic products globally, microplastics have become ubiquitous and persistent pollutants in the environment. Recent studies indicated that microplastic pollution levels in terrestrial ecosystems might be dozens of times higher than those in marine ecosystems [5,6]. It is estimated that 473,000 to 910,000 metric tons of plastic waste remain in terrestrial environments within the European Union each year, equivalent to 4 to 23 times the total amount discharged into marine environments during the same period [7]. A research study that analyzed soil aggregates in Southwest China revealed an average microplastic abundance of 18,760 particles per kilogram [8]. Among various ecosystems, agricultural ecosystems are among those most severely affected by microplastic pollution [9]. Microplastic abundance in the soils of China’s major agricultural lands could reach 2783 to 6366 particles/kg [10]. Specifically, the abundance of microplastics in the soils of suburban vegetable-growing areas in Wuhan, China, ranges from 320 to 12,560 particles/kg [11]. Along the lower reaches of the Yangtze River, the abundance of microplastics in farmland soils ranges from 4.94 to 252.70 particles/kg [12]. In addition, in Yakehekou Reservoir (China), the primary types of microplastics are polypropylene (29.41%), polyethylene (38.89%), polyethylene terephthalate (PET) (10.13%), and polyvinyl chloride (PVC) (12.09%) [13]. In the 0~30 cm soil layer of farmland, a large number of microplastic particles, such as polyethylene (PE), polystyrene (PS) and polypropylene (PP) microplastics have been detected [14,15]. Consequently, mounting concern has been directed towards the potential ecological risks posed by high concentrations of microplastics in the soil, which affect the environment through the soil–plant system [16,17].

Microplastics, due to their difficult-to-degrade nature [18], remain in soil environments for extended periods after entering terrestrial ecosystems, exerting significant impacts on soil–plant systems [9]. Their persistent retention can alter soil physicochemical properties and affect the diversity and community structure of soil animals, plants, and microorganisms [19]. Studies have confirmed that microplastics interact with soil and indirectly affect plant growth [20,21]. Treatment with 0.02% polystyrene led to a significant decrease in chlorophyll a, chlorophyll b, and total chlorophyll content in wheat seedlings [22]. Exposure to microplastics alone resulted in a reduction in maize seedling biomass and water content [23]. Furthermore, the SOD and CAT activities in rice were found to be significantly inhibited by 0.1% polystyrene treatment [24]. Specifically, microplastics may affect the absorption and transport of nutrients such as potassium (K) and iron (Fe) [25]. This occurs because microplastic particles accumulate in root tissues and pores in cell walls may become blocked, thereby preventing plants from absorbing nutrients and water [26]. In addition, microplastics adhere not only to root surfaces but also to pores in seed coats, potentially causing mechanical damage to plant roots and thereby affecting plant growth [9]. In recent years, as agricultural microplastic pollution has garnered increasing attention, research on the phytotoxicity of microplastics has primarily focused on vegetable and crop plants, including wheat (*Triticum aestivum* L.) [27,28], corn (*Zea mays* L.) [29], tomato (*Lycopersicon esculentum* L.) [19,30], cucumber (*Cucumis sativus* L.) [31], and lettuce (*Lactuca sativa* L.) [32,33]. However, to date, few studies are available on the toxic effects of microplastics on oats (*Avena sativa* L.), an important feed crop.

Forage growth is fundamental to animal husbandry and is directly linked to animal health and productivity. The impact of microplastic pollution on feed crops is crucial not only for the sustainable development of artificial grasslands and farmlands but also for its potential implications for human health via the food chain. In China, oats are an adaptable, high-yield crop used for grain, feed and dual purposes. Oats serve as a promising alternative energy source with balanced nutritional properties. They are an excellent source of carbohydrates and high-quality protein, featuring a favorable amino acid profile. Oats are rich in lipids, particularly unsaturated fatty acids, as well as minerals, vitamins, and phytochemicals [34]. In China, the oat cultivation area is approximately 525,000 hectares, with an annual production of about 8.5 million tons. The planting area is expanding annually to address the tight feed supply situation. Given their significance in the feed industry, oats represent an ideal model for assessing the effects of microplastic pollution on feed crops.

Although numerous studies have demonstrated the harmful effects of individual microplastic polymers on plants [23], the impact of diverse microplastic polymers on feed crop toxicity remains an under-researched area. Therefore, we investigated whether soil microplastic pollution adversely affected feed crops through experiments on microplastic contamination in oats. We hypothesized that microplastics exerted toxic effects on oat growth and development, and that the toxicity varied with microplastic types and concentrations. For this study, the microplastics chosen were two commonly recognized types detected in soil: polyethylene (PE) and polystyrene (PS) microplastics [35,36]. The aim of this study was to explore the impacts of microplastics on oat seedling growth and physiological processes, and to ascertain whether specific types and concentrations of microplastics had distinct effects on oat seedlings. This research offers theoretical underpinnings and references for evaluating the safety of microplastics within agricultural ecosystems and artificial grasslands.

## 2. Results

### 2.1. Morphological Characteristics of Microplastics

Figure 1 revealed that PS microplastics exhibited spherical morphology, whereas PE microplastics displayed irregular surface structures characterized by sharp edges.

### 2.2. Effects of Microplastics on Oat Seedling Growth

All treatments caused significant decreases in plant height, root length, root fresh weight, stem diameter and leaf area compared to the Ctrl group (*p* < 0.05), except for plant height under 0.1% PS microplastic treatment (Figure 2). Specifically, plant height decreased by 18.4% to 31.1% under 0.5% to 5% PS microplastic treatment, and by 14.2% to 40.8% under 0.1% to 5% PE microplastic treatment compared to the Ctrl group, respectively. Interestingly, the root length did not decrease significantly with an increase in PS microplastic concentration (*p* > 0.05). On the contrary, with 0.1~5% PE microplastics, the root length was reduced by 18.7% to 44.6%. The root fresh weight of oats was significantly inhibited by 0.1~5% PS and PE microplastic treatments (*p* < 0.05), with inhibition rates of 70.7% to 60.6% and 43.7% to 80.9%, respectively, suggesting a greater impact of microplastics on root fresh weight than on root length. Under 0.1~5% PS and PE exposure conditions, the inhibition rates for stem diameter were 17.9% to 39.2% and 16.3% to 44.9%, respectively. The leaf area was also inhibited, with reduction rates of 45.3% to 71.9% for PS and 28.1% to 77.4% for PE microplastics. At the 5% treatment concentration, PS and PE both had the strongest inhibitory effect on seedling growth, and PE microplastics exhibited larger effects on oat seedling growth indicators (root length, plant height, and root fresh weight) than PS microplastics.

### 2.3. Effects of Microplastics on Chlorophyll Content in Oat Leaves

PS microplastics showed strong inhibition of chlorophyll a, chlorophyll b, and total chlorophyll at 1% and 5% concentrations (*p* < 0.05), while PE microplastics significantly inhibited chlorophyll a, chlorophyll b, and total chlorophyll at 0.5%, 1%, and 5% concentrations when compared to the Ctrl group (*p* < 0.05). However, chlorophyll a, chlorophyll b, and total chlorophyll under the 1% PS microplastic treatment did not differ significantly from those under the 5% PS microplastic treatment (*p* > 0.05), suggesting a complex impact of PS microplastics on chlorophyll content in oats. Moreover, PE microplastics showed stronger inhibitory effects on chlorophyll a, chlorophyll b and total chlorophyll than PS microplastics at the same concentrations (0.5% and 5%), with inhibition intensifying as concentrations increased (*p* < 0.05). Chlorophyll a, chlorophyll b, and total chlorophyll were the lowest in the 5% PE microplastic treatment, decreasing by 61.4%, 37.6%, and 61.4%, respectively, compared to the Ctrl group (Figure 3).

### 2.4. Effects of Microplastics on Photosynthetic Parameters in Oat Leaves

As presented in Figure 4, the PS microplastic treatment significantly enhanced the net photosynthetic rate of oats by 14.0% at a concentration of 0.1% (*p* < 0.05), compared to the Ctrl group. The 1% and 5% PE microplastic treatments significantly decreased net photosynthetic rates by 25.3% and 45.2% (*p* < 0.05), respectively, with the 5% concentration reaching the minimum value. The stomatal conductance of oats exhibited negative responses to both PS and PE microplastic treatments, showing a gradual decrease with increasing concentrations. Compared to the Ctrl group, stomatal conductance under PS microplastic treatments decreased by 38.4%, 67.5%, and 71.1% at concentrations of 0.5%, 1%, and 5%, respectively; under PE microplastic treatment, the stomatal conductance decreased by 53.3%, 69.7%, and 90.5%, respectively. Compared with the Ctrl group, the intercellular CO_2_ concentration decreased by 43.1% under the 5% PS microplastic treatment; however, the 5% PE microplastic treatment had a significantly stronger inhibitory effect on this parameter (*p* < 0.05). The 0.5% and 1% PS microplastic treatments significantly reduced transpiration rates by 18.9% and 22.1%, compared to the Ctrl group (*p* < 0.05). At the same concentrations (0.5% and 1%), PE microplastic treatments exhibited a more significant inhibitory effect on the transpiration rate of oats compared with PS microplastic treatments (*p* < 0.05). Specifically, the transpiration rate was reduced by 38.3% and 41.9% under 0.5% and 1% concentrations in PE microplastic treatment groups; the lowest transpiration rate was observed in the 5% PE microplastic treatment group.

### 2.5. Effects of Microplastics on Antioxidant Enzyme Activities in Oats

Both PE and PS microplastics significantly altered the POD and CAT activities in oats (*p* < 0.05). POD activity was improved following 0.5~1% PS and 0.1~1% PE microplastic treatments and then decreased following 5% PS and 5% PE microplastic treatments, although there was no significant difference between the 5% PS microplastic treatment and the Ctrl group (*p* > 0.05). Similarly to POD activity, CAT activity significantly increased following 0.5~1% PS and decreased following 5% PS microplastic treatment (*p* < 0.05), whereas it was significantly inhibited following 1% PE and 5% PE microplastic treatments. Therefore, POD and CAT activities were promoted by low and medium concentrations but inhibited by high concentrations. Moreover, SOD activity was less responsive compared with POD and CAT activities when exposed to microplastics. Among all treatments, significant decreases in SOD activity were noted only at 5% PS and 5% PE microplastic treatments when compared to the Ctrl group (*p* < 0.05). Additionally, a clear trend was observed in which MDA content increased progressively with increasing concentrations under microplastics exposure. The MDA content increased by 101.7% and 190.8% under 5% PS and 5% PE microplastic treatments, respectively. At the same concentration (5%), the PE microplastic treatment exhibited a stronger inducing effect on MDA content compared to the PS microplastic treatment (Figure 5).

### 2.6. Interaction Effects of Microplastic Types and Concentrations on Oats

All indicators of oats, except for SOD activity, were significantly affected by microplastic concentration (*p* < 0.05). Microplastic type significantly influenced Chla, Chlb, Chl, Pn, Tr, LWC, PH, SOD, CAT, and MDA (*p* < 0.01), and Gs, and LA (*p* < 0.05). Chla, Chlb, Chl, RL, RW, SD, LWC, PH, and CAT (*p* < 0.01) and Ci, LA, Pn, and MDA (*p* < 0.05) were all significantly influenced by the Type×Concentration interaction effect (Table 1). This indicated that the interaction effects of microplastic type and concentrations on these indicators were variable. A significant correlation existed between microplastic type and concentration, with microplastic concentrations exerting a stronger impact on oats (*p* < 0.05).

We observed that PC1 retained 63.2% of the data variation, while PC2 retained 12.2% (Figure 6). The values of Gs, Chla and Chlb, PH and POD in oats were significantly correlated with the first main axis (*p* < 0.05), indicating that these indicators were the key factors driving oats’ responses to different microplastic treatments.

## 3. Discussion

This study investigated the effects of microplastic pollution on oat growth and development and explored the underlying mechanisms. Microplastics might affect plants indirectly by altering soil physicochemical properties and microbial community structure; however, this study focused primarily on the direct physiological effects. Our results demonstrated that the phytotoxicity of microplastics depended on both their type and concentration. Scanning electron microscopy (Figure 1) revealed distinct surface characteristics: PS microplastics showed a relatively smooth surface, whereas PE microplastics exhibited an irregular morphology with sharp edges. This structural difference might explain the stronger inhibitory effect of PE microplastics on oat growth. The irregular edges of PE microplastics could mechanically damage root cell walls upon uptake, causing direct physical injury. This mechanism was consistent with reports that irregularly shaped microplastics more severely affected plant biomass and soil microbial communities [5]. While earlier studies also found PE microplastics were more inhibitory than PS microplastics in tomatoes [19] and in wheat [28], our findings directly linked the greater toxicity of PE microplastics to their potential for mechanical damage, as evidenced by the correlation between microplastic morphology (Figure 1) and the magnitude of growth reduction in our experiments. Concentration was another critical factor. Generally, higher microplastic concentrations were associated with more pronounced inhibition of plant growth [37]. Our results showed that exposure to 1% and 5% PE microplastics significantly reduced oat plant height, root length, root fresh weight, stem diameter, and leaf area. These findings were consistent with previous studies reporting that high concentrations (2%) of PE microplastics decreased biomass accumulation in camphor seedlings [38]. Urbina et al. (2020) [39] suggested that the inhibitory effects at elevated concentrations were primarily attributable to the physical obstruction of root surfaces. Our results supported this mechanism, as the observed suppression of root elongation under the 5% PE treatment was indicative of pore blockage and impaired water uptake. Notably, certain studies reported that very high concentrations (8% PE microplastics, 200 μm) did not inhibit wheat root growth, likely because larger particles were unable to penetrate cell walls [27,40]. Therefore, the more pronounced inhibitory effect of PE microplastics on oat growth could be primarily attributed to physical blockage and mechanical injury caused by high concentrations of PE microplastics, which disrupted root function and impaired physiological processes, ultimately suppressing normal plant development.

Chlorophyll content is a key indicator of plant photosynthetic capacity, indirectly reflecting the plant’s current growth status. In this study, the damage to the chlorophyll indicators of oat leaves gradually intensified with the increase in PE microplastic concentration, indicating a concentration-dependent relationship between PE microplastic exposure and chlorophyll content changes. Cucumber seedlings exhibited a gradual decrease in chlorophyll content as the concentration of PE microplastics rose from 0.004% to 0.1% [31]. The chlorophyll content in wheat seedlings was not affected by low concentrations of microplastics, while a significant decrease was observed at higher concentrations [27]. Chlorophyll levels, a key factor determining seedling height, remained largely unchanged in oats under the 0.1% PS microplastic treatment. This could be attributed to plant defense mechanisms under environmental stress. One possible explanation was that plants accelerated chlorophyll synthesis to increase its content, thereby enhancing the efficiency of the photosynthetic system and better adapting to adverse environmental conditions [41]. This stability in chlorophyll content facilitated normal photosynthetic activity, which could explain the observed promotion of plant height. These findings confirmed that oats could tolerate the mild stress induced by 0.1% PS microplastics and sustain growth. Previous studies had shown that *Pinus* and *Sequoia* enhanced photosynthesis by increasing chlorophyll content to resist low concentration microplastic stress and maintain growth [42]. Conversely, high concentrations (5%) of microplastics significantly inhibited chlorophyll content in oat leaves. This might be due to high-level stress inducing structural changes or even disintegration of certain pigment proteins in the chloroplasts, which in turn altered the activity of chlorophyll-synthesizing enzymes, ultimately reducing chlorophyll content and impairing plant growth [41]. Li et al. (2020) [43] further noted that plants exposed to microplastic stress produced large amounts of reactive oxygen species, which disrupted chlorophyll structure. Our study supported this observation: under treatment with a high concentration (5%) of microplastics, a significant decrease in chlorophyll content in oat leaves was observed, while the MDA content increased sharply. The key mechanism underlying these changes may be a reactive oxygen species (ROS) burst induced by microplastic stress triggering severe membrane lipid peroxidation. This process likely disrupted chlorophyll synthesis, ultimately leading to chlorophyll decomposition and a decline in photosynthetic capacity. In summary, the changes in the photosynthetic pigment content of oat leaves were closely related to soil microplastic pollution, with the magnitudes and trends of these variations depending on microplastic type and concentration.

In addition, plant growth and development were critically regulated by photosynthesis, and environmental stress significantly suppresses photosynthetic efficiency [44]. Directly, microplastics attached to root surfaces and interfered with essential root functions, consequently disrupting photosynthetic processes. Moreover, microplastics could act as carriers for co-existing pollutants in the soil. For instance, they could form complexes with heavy metals like cadmium and lead [45]. This interaction altered the bioavailability of these pollutants, thereby indirectly exacerbating the reduction in photosynthetic efficiency [46,47,48,49]. Consistent with these mechanisms, it was reported that water hyacinth exhibited significantly inhibited Pn after 48 h of treatment with PS microplastics [50]. Microplastics adhering to the surfaces of lettuce leaves could impede light transmission, thereby hindering normal photosynthesis [51]. Additionally, Zhao et al. (2017) [52] and Zhang et al. (2022) [29] revealed that substantial accumulation of PS microplastics in plant leaf stomata led to a reduction in stomatal Gs values. In this study, both PS and PE microplastics significantly reduced Gs values under high concentrations (5%). Studies on Masson pine reported that when net photosynthetic rates decreased and intercellular CO_2_ concentrations increased, the primary factor inhibiting plant photosynthesis was non-stomatal limitation [53]. However, stomatal conductance and intercellular CO_2_ concentrations decreased simultaneously, indicating that the limitation of photosynthesis was due to the stomatal factor. Our research revealed that high concentrations (5%) of PE and PS microplastics significantly inhibited both stomatal conductance and intercellular CO_2_ concentration. This occurred because high microplastic concentrations caused stomatal closure in plants, thereby inducing stomatal limitation of photosynthesis [54]. Additionally, stomata closure indirectly affected transpiration in leaves [50]. We proposed that stomatal closure was a systemic response of plants to root stress. This was consistent with the findings of this study, which suggested that the accumulation of microplastics in the rhizosphere caused physical blockage and mechanical damage. This might interfere with water and nutrient absorption and triggered root stress, ultimately inducing stomatal closure to reduce water loss.

Under abiotic stress, plants initiated oxidative stress responses, which triggered the excessive accumulation of reactive oxygen species (ROS) in cells and subsequently led to oxidative damage to biomolecules [55]. Changes in the activity of antioxidant enzymes were widely used to assess the severity of external stress exposure in plants [56]. Soil microplastics could physically obstruct plant roots and cause mechanical damage at the cellular level, disrupting plant homeostasis and inducing a state of stress, thereby interfering with antioxidant scavenging capacity [57]. Under mild stress, low concentrations of ROS acted as signaling molecules to enhance antioxidant capacity, thereby counteracting excessive ROS accumulation [58]. This study found that low-concentration (0.1%, 0.5%, and 1%) PE treatment significantly promoted POD activity, while PS treatment (0.5% and 1%) significantly enhanced the activity of both CAT and POD enzymes. Liu et al. (2021) [27] observed that 1% PE microplastics significantly increased antioxidant enzyme activity in wheat seedlings. We proposed that low concentrations of microplastics exerted mild stress on oats, inducing oxidative stress adaptation in seedlings [42]. Consistent with this mechanism, the results of this study indicated that microplastic stress disrupted the redox homeostasis of oats, leading to the accumulation of ROS. In response, under low concentration stress, adaptive defense was achieved by enhancing antioxidant enzyme activity. In contrast, as environmental stress intensified, excessive ROS accumulation occurred, weakening the plant’s antioxidant defense capacity, disrupting redox homeostasis, and ultimately resulting in decreased antioxidant enzyme activity [59,60]. In our study, oats treated with 5% PE microplastics exhibited significant reductions in POD, CAT, and SOD activities. The decline likely resulted from increased microplastic concentrations exacerbating physical blockage and mechanical damage, thereby impairing plant antioxidant capacity and disrupting its self-regulatory capacity, ultimately causing severe oxidative damage [61]. Similarly, microplastic exposure significantly reduced antioxidant enzyme activity and markedly suppressed the synthesis of related proteins in tomatoes. Consequently, these effects led to excessive ROS accumulation and oxidative damage [19,24]. Specifically, under high concentration stress, although ROS continued to accumulate, the activity of antioxidant enzymes decreased significantly. The above phenomenon could be explained by the following mechanism: excessive stress overwhelmed the plant‘s enzymatic antioxidant capacity, which led to a complete imbalance in redox homeostasis and caused severe oxidative damage. This was evidenced by the sharp increase in MDA content observed in this experiment. Our study found that 5% PE microplastics exerted a significantly stronger effect on the POD and CAT activities of oats than 5% PS microplastics. Different types of microplastics exhibited differential toxic effects on plants [62,63,64]. At the same concentration, PE microplastics might exert a more pronounced effect on the antioxidant enzyme activity of seedlings than PS microplastics [42]. This phenomenon could be attributed to differences in the selective adsorption of microplastics by plants, as well as variations in the degradability of microplastics and the toxicity of their degradation products.

The defense mechanisms of plants against microplastic stress involved multiple pathways. While this study primarily focused on the activities of antioxidant enzymes, secondary metabolites (such as flavonoids and phenolic compounds) played an equally crucial role in scavenging reactive oxygen species (ROS) and protecting plants from oxidative stress induced by microplastics [65,66]. These secondary metabolites effectively reduced oxidative damage through their free radical scavenging abilities and metal chelating properties, thereby providing an additional protective layer. Flavonoids were essential non-enzymatic metabolites that enhanced plant stress tolerance [67]. Previous studies showed that microplastic stress affected the accumulation of flavonoids, which served as a protective mechanism against oxidative stress [68]. Specifically, PS microplastic treatment significantly enhanced flavonoid accumulation in dandelion seedlings (*Taraxacum mongolicum* Hand.-Mazz.) [69]. Under nano-plastics (NPs) stress, genes involved in flavonoid biosynthesis were significantly upregulated in lettuce [70]. This suggested that mild stress might promote the biosynthesis of flavonoids, enhancing the plant’s ability to alleviate excessive oxidative stress induced by microplastics by eliminating excess ROS and protecting plant cells. In the present study, 0.5% PS microplastic treatment significantly enhanced antioxidant enzyme activity, demonstrating that mild stress strengthened plant antioxidant defenses. Concomitantly, flavonoid accumulation might also have been stimulated. Enzymatic antioxidants and flavonoid metabolites scavenged excessive ROS through a synergistic action. Conversely, under PET microplastic treatment, severe oxidative damage was induced in pepper seedlings (*Capsicum annuum* L.), ultimately suppressing flavonoid biosynthesis and reducing flavonoid accumulation [71]. This could be attributed to the toxic effects of PET, which inhibited the synthesis of secondary metabolites in plants and consequently induced severe oxidative damage. In this study, the significant inhibition of antioxidant enzyme activities under high-concentration (5%) PE microplastic treatment and the sharp increase in MDA content jointly indicated that severe oxidative damage occurred. This phenomenon suggested that extreme stress could weaken the overall antioxidant defense capacity of plants; consequently, the content of secondary metabolites such as flavonoids, which were a key link in this process, might also be significantly inhibited, thereby impairing the ROS scavenging capacity of plants.

Furthermore, under adverse environmental stress, plants generated oxygen free radicals through both enzymatic and non-enzymatic systems. These radicals attacked polyunsaturated fatty acids in cell membranes, triggering lipid peroxidation and producing MDA. This process served as an indicator of the plant’s stress response, with the content of MDA being a crucial metric for assessing the severity of stress experienced by the plant. The findings of this study demonstrated that treatments with 5% PS, 0.5% PE, 1% PE, and 5% PE microplastics all significantly increased MDA content in oats. These results confirmed that microplastic stress induced severe oxidative damage to the cell membranes of oats.

Therefore, our findings revealed that the growth inhibition in oats under microplastic stress was primarily driven by a cascading mechanism: the uptake and accumulation of microplastics in the root system caused mechanical damage and physical blockage, disrupting redox homeostasis and triggering oxidative damage. This oxidative impairment subsequently hampered normal photosynthesis by inhibiting the synthesis of photosynthetic pigments, thereby ultimately suppressing plant growth and development.

In summary, this study demonstrated that microplastic concentration was the key factor affecting oat growth, whereas microplastic type only exerted significant impacts on a limited number of indicators. Both PS and PE microplastics exerted inhibitory effects on oats, as evidenced by growth, photosynthetic, and antioxidant parameters. Notably, the inhibitory effect of PE microplastics on oats was concentration-dependent, as demonstrated by substantial reductions in seedling growth, chlorophyll content, photosynthetic parameters, and antioxidant enzyme activity with increasing microplastic concentrations. However, oats exhibited a significant trade-off between growth and stress responses at different microplastic concentrations. Under adverse conditions, plants reallocated resources from growth to stress responses, thereby affecting developmental processes [72]. When the resources required for stress responses exceed a specified threshold, plants might suppress growth to ensure their own survival. Consequently, under microplastic stress, management practices such as fertilization could be employed to optimize resource allocation and alleviate seedling growth suppression, thus preventing yield losses [73]. The concentration gradient established in this study fell within the upper range of the reported field pollution levels, reflecting the actual microplastic contamination levels found in farmland soil. Therefore, the results from the 0.1%, 0.5%, and 1% treatments could be directly applied to assess ecological risks in currently contaminated areas. In contrast, the high-concentration (5%) treatment helped us to understand potential worst-case scenarios that potential long-term effects might be caused and predict critical thresholds for microplastic pollution. Specifically, it was recommended that fertilizer supplementation could be appropriately increased in areas where levels of pollution were high to alleviate stress and ensure optimal crop growth. In addition, regarding feed safety, the uptake and translocation of microplastics in forage crops such as oats could pose a potential risk to livestock health. Therefore, systematically assessing their transfer through the feed chain was crucial to safeguarding animal health and, ultimately, the safety of animal products. From a better perspective, our data on concentration-dependent effects provided a scientific basis for developing soil microplastic threshold guidelines and highlighted the need to include microplastic pollution in the monitoring and assessment framework of agricultural ecosystems. However, it is important to note the limitations of this pot experiment. The controlled environment, while ideal for elucidating mechanistic pathways, might not fully replicate the complexity of field conditions. Therefore, future research should prioritize long-term field trials to assess the ecological risks of microplastics.

## 4. Materials and Methods

### 4.1. Experimental Materials

The pot experiments were conducted in Luoyang City, Henan Province, China. Luoyang City has a warm temperate continental monsoon climate, featuring four distinct seasons, a mild and humid climate, abundant sunshine, and ample rainfall. The oat variety selected for the pot experiments was “Sweet Oat 60,” which was purchased from the Luoyang Academy of Agricultural Sciences. This variety had a high digestible fiber content and strong adaptability. The yellow fluvo-aquic soil employed for the pot experiments was collected from the farm of Henan University of Science and Technology. The basic physicochemical properties of the yellow fluvo-aquic soil are as follows: the pH level is 7.96, the organic carbon content is 10.91 g/kg, the alkali-hydrolyzable nitrogen content is 79.03 mg/kg, the available phosphorus content is 11.37 mg/kg, the readily available potassium content is 226.8 mg/kg, and the available iron content is 6.18 mg/kg. No microplastic residues were detected in the collected soil samples. It was then dried outdoors and sieved through a 2 mm sieve before use.

Research surveys showed that the most common types of microplastics in Chinese farmland and grassland soil were PE and PS microplastics [74,75], and most microplastics were less than 10 μm in diameter [76]. Therefore, we selected 2 μm PE and PS microplastics as test materials to simulate microplastic pollution in farmland soil. All microplastics used in this study were primary microplastics, sourced directly as commercial 3D printing powders without additives from Dongguan Zhangmutou Suzhan Plastic Products Trading Department. The purchased microplastics were scanned using SEM to analyze the morphological characteristics of the two types of microplastics. Microplastic concentrations were set at 0%, 0.1%, 0.5%, 1%, and 5% (*w*/*w*) of soil mass, with no microplastics added as the Ctrl group (Ctrl). According to previous studies by Li et al. (2024) [42], Liu et al. (2021) [27] and Machado et al. (2018) [77], these concentrations (0.1%, 0.5%, 1%) reflected actual soil pollution levels. Given the increasingly severe plastic pollution worldwide [78], using a high concentration (5%) allows for predicting the worst-case scenario of microplastic pollution and magnifying potential side effects, which might otherwise be overlooked. The microplastics of different concentrations were evenly mixed with the soil in the pots and left to stand for 10 days. Uniform-sized seeds were selected and soaked in 2% hydrogen peroxide (H_2_O_2_) solution for 30 min. Then, they were repeatedly washed with ultrapure water to ensure the removal of the disinfectant. After that, they were sown in the prepared pots, with 20 seeds per pot, and each treatment was repeated five times. Therefore, the pot experiments had 9 treatment groups (Ctrl, 0.1% PS, 0.5% PS, 1% PS, 5% PS and 0.1% PE, 0.5% PE, 1% PE, 5% PE), using a completely randomized block design, with a total of 45 pots. The pots used for growing oats were ceramic, with a diameter of 20.5 cm, a height of 18.5 cm, a base diameter of 14.5 cm, and were filled with 4 kg of soil. The pot experiments for oat cultivation were conducted outdoors. The pot experiments began on 30 September 2024 and lasted until 30 November 2024. Oats remained in their growing phase throughout this period and did not enter dormancy. The average temperature during the cultivation period was 18 °C, and the average rainfall was 152.3 mm. On non-rainy days, artificial watering was provided to the seedlings to ensure adequate water supply and weeds were removed from the pots.

### 4.2. Experimental Methods

#### 4.2.1. Determination of Morphological Indicators in Oats

After 60 days of growth, the plant height and stem diameter were measured using a tape measure and a vernier caliper, respectively. The first intact leaf from the top was selected, and its length and width were measured to calculate the leaf area. The leaf area was calculated using the following formula:A(leaf) = L × W × 0.76
where L represents the unfolded length of the oat leaves, and W represents the unfolded width of the oat leaves. After the oats were harvested, the root length was measured using a tape measure, and the fresh weight of the roots was determined using a 0.0001 g precision balance.

#### 4.2.2. Determination of Chlorophyll Content in Oats

Fully mature leaves were collected from the same position on each oat plant at 60 days after planting for chlorophyll content measurement. The leaves were washed thoroughly, and their surfaces were blotted dry with absorbent paper. Next, 0.1 g of fresh leaves was weighed, chopped finely, and placed in a 15 mL centrifuge tube. Then, 10 mL of 95% ethanol was added, the tube was covered, and it was left to stand in the dark for 24 h until the leaves had completely lost their green color. A 95% ethanol solution was used as the blank Ctrl group. After the extract was shaken thoroughly, its absorbance was measured at wavelengths of 665 nm and 649 nm. The chlorophyll a, chlorophyll b, and total chlorophyll contents were calculated according to established equations [79].Chla content (μg/mL) = 13.95 OD665 − 6.88 OD649(1)Chlb content (μg/mL) = 24.96 OD649 − 7.32 OD665(2)Chlorophyll content = (Chla or Chlb content × extraction liquid volume × dilution factor)/fresh weight of sample (g)(3)

#### 4.2.3. Measurement of Gas Exchange Parameters and Photosynthetic Rates in Oats

After 45 days of growth, fully expanded mature oat leaves were selected from the same plant positions on sunny mornings between 9:00 and 11:00. The net photosynthetic rate (Pn), stomatal conductance (Gs), intercellular CO_2_ concentration (Ci), and transpiration rate (Tr) of these leaves were then measured using a LI-6800 portable photosynthesis system (LI-COR, Inc., Lincoln, NE, USA). Prior to measurement, the airtight integrity of the leaf chamber was ensured. The light intensity was set to 1000 μmol m^−2^ s^−1^. After the readings stabilized, five readings were recorded. Following the removal of outliers, the mean value was calculated.

#### 4.2.4. Determination of Antioxidant Enzyme Activity in Oats

The oat root tissue was weighed to 0.05 g and placed in a 5 mL centrifuge tube. Then, 0.45 mL of phosphate-buffered saline (PBS) was added. The tissue was homogenized using a handheld grinder under an ice-water bath. The homogenate was centrifuged at 6000 rpm for 20 min. The supernatant was collected, and the activities of total superoxide dismutase (SOD), catalase (CAT), and peroxidase (POD) were determined according to the instructions of the assay kit (Nanjing Jian Cheng, Nanjing, China). The absorbance of the antioxidant enzymes was measured at specific wavelengths using an enzyme-linked immunosorbent assay reader.

#### 4.2.5. Assessment of Membrane Peroxidation in Oats

When plant tissues sustain damage under stress conditions, lipid peroxidation occurs. Malondialdehyde (MDA) is the final breakdown product of lipid peroxidation, and its concentration reflects the severity of stress-induced damage in plants. The malondialdehyde (MDA) content was determined following the procedure outlined in Section 4.2.4, with the modification that the reagent added prior to grinding was replaced with the extraction solution provided in the kit (Nanjing Jian Cheng, Nanjing, China).

### 4.3. Data Analysis

Experimental data were processed and statistically analyzed using Excel 2012 software. One-way ANOVA and Duncan’s multiple range test (*p* < 0.05) were performed on the data using SPSS 25.0 software. All experimental results were expressed as the mean ± standard error (mean ± SE). The effects of different types and concentrations of microplastics on various indicators of oats were analyzed using a generalized linear model. Principal component analysis (PCA) was employed to analyze the correlations among different seedling indices. GraphPad Prism 8.0 and Origin 2021 software were used for data plotting.

## 5. Conclusions

The present study investigated the effects of PS and PE microplastics on the growth, photosynthetic, and antioxidant parameters of oats. The results revealed that the inhibitory effect of PE microplastics on seedling growth was concentration-dependent, and gradually intensified with increasing PE microplastic concentration. Conversely, PS microplastics demonstrated a more complex concentration–response relationship, exhibiting distinct responses to seedling growth under varying concentration treatments. Furthermore, at the same concentration (5%), PE microplastics demonstrated a more pronounced inhibitory effect on oats than PS microplastics. Our research demonstrated that microplastic accumulation in roots initiated a cascade: physical damage and blockage first disrupted redox homeostasis, triggering oxidative stress. This oxidative stress then impaired photosynthesis by inhibiting photosynthetic pigment synthesis, leading ultimately to suppressed plant growth and development.

## Figures and Tables

**Figure 1 plants-15-00056-f001:**
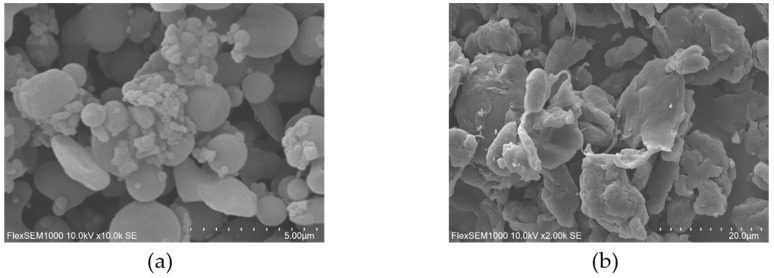
Microplastics characterization. Scanning electron microscopy (SEM) images of PS (**a**) and PE (**b**).

**Figure 2 plants-15-00056-f002:**
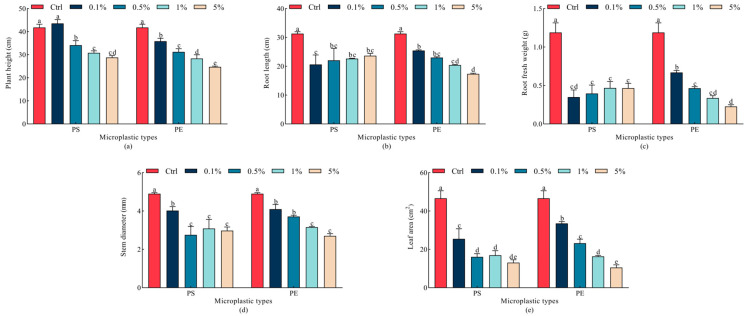
Changes in growth parameters in oat seedlings under different microplastic treatments. (**a**) Plant height, (**b**) Root length, (**c**) Root fresh weight, (**d**) Stem diameter, (**e**) Leaf area. All data are expressed as means ± SD (*n* = 5 biological replicates). A one-way ANOVA was conducted, followed by a Duncan’s post hoc test. Lowercase letters above bars indicate significant differences (*p* < 0.05). Ctrl, control group; PE, polyethylene; PS, polystyrene.

**Figure 3 plants-15-00056-f003:**
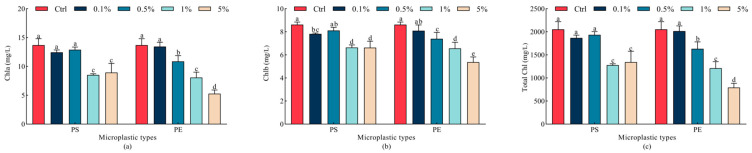
Changes in chlorophyll content in oat leaves under different microplastic treatments. (**a**) Chla, (**b**) Chlb, (**c**) Total Chl. All data are expressed as means ± SD (*n* = 5 biological replicates). A one-way ANOVA was conducted, followed by a Duncan’s post hoc test. Lowercase letters above bars indicate significant differences (*p* < 0.05). Chla, chlorophyll a; Chlb, chlorophyll b; Total Chl, total chlorophyll; Ctrl, control group; PE, polyethylene; PS, polystyrene.

**Figure 4 plants-15-00056-f004:**
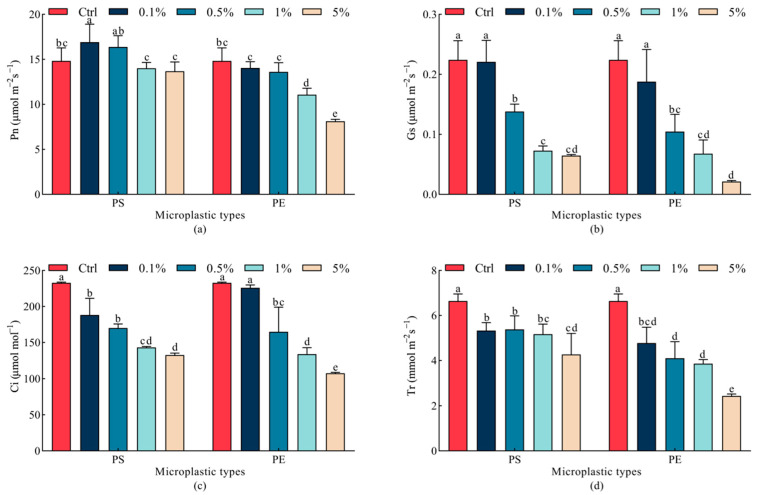
Changes in photosynthetic parameters of oat leaves under different microplastic treatments. (**a**) Pn, (**b**) Gs, (**c**) Ci, (**d**) Tr. All data are expressed as means ± SD (*n* = 5 biological replicates). A one-way ANOVA was conducted, followed by a Duncan’s post hoc test. Lowercase letters above bars indicate significant differences (*p* < 0.05). Pn, net photosynthetic rate; Gs, stomatal conductance; Ci, intercellular CO_2_ concentration; Tr, transpiration rate; Ctrl, control group; PE, polyethylene; PS, polystyrene.

**Figure 5 plants-15-00056-f005:**
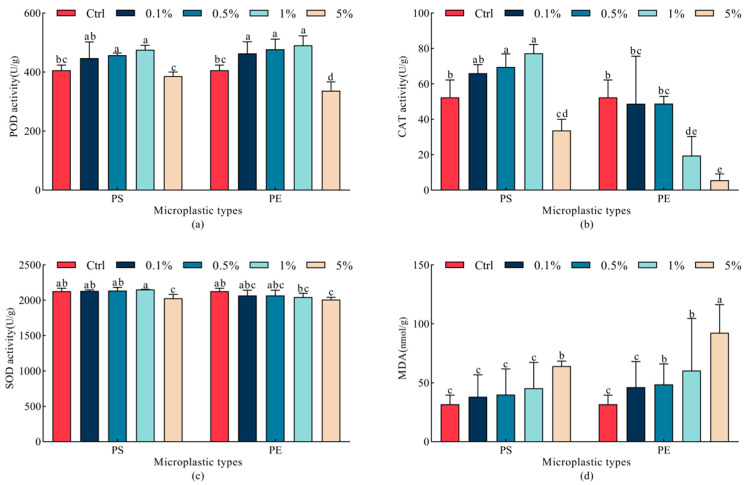
Changes in antioxidant enzyme activities of oat leaves under different microplastic treatments. (**a**) POD activity, (**b**) CAT activity, (**c**) SOD activity, (**d**) MDA content. All data are expressed as means ± SD (*n* = 5 biological replicates). A one-way ANOVA was conducted, followed by a Duncan’s post hoc test. Lowercase letters above bars indicate significant differences (*p* < 0.05). POD, peroxidase; CAT, catalase; SOD, superoxide dismutase; MDA, malondialdehyde; Ctrl, control group; PE, polyethylene; PS, polystyrene.

**Figure 6 plants-15-00056-f006:**
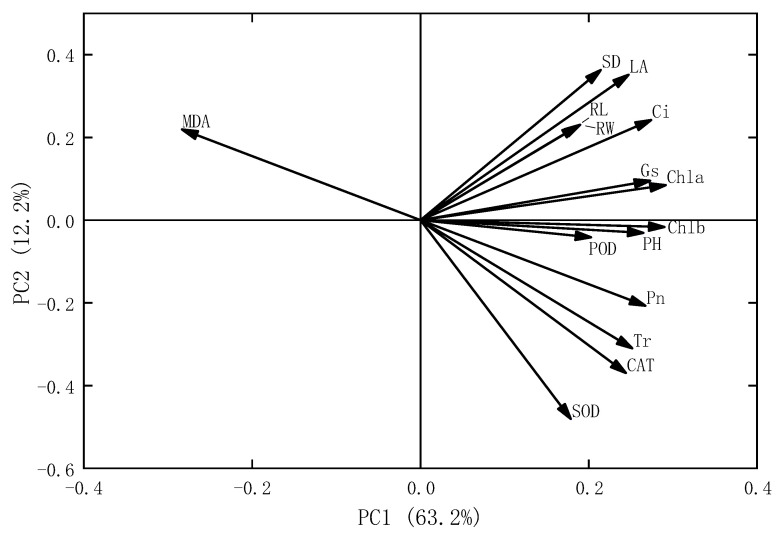
Principal component analysis of the growth and physiological characteristics of oats under different microplastic treatments. Note: The horizontal and vertical axes represent the first and second principal components of PCA, respectively. Chla, chlorophyll a; Chlb, chlorophyll b; Pn, net photosynthetic rate; Gs, stomatal conductance; Ci, intercellular CO_2_ concentration; Tr, transpiration rate; RL, root length; SD, stem diameter; LA, leaf area; PH, plant height; POD, peroxidase; SOD, superoxide dismutase; CAT, catalase; MDA, malondialdehyde.

**Table 1 plants-15-00056-t001:** Effects of microplastic types, concentrations and their interactions on growth and physiological indicators of oats.

Indicator	Parameter	Mps Type	Mps Concentration	Type × Concentration
		Significance Level (*p*)		
Growth indicators	PH	0.000 **	0.000 **	0.002 **
	RL	0.401	0.000 **	0.000 **
	RFW	0.898	0.000 **	0.000 **
	SD	0.076	0.000 **	0.003 **
	LA	0.029 *	0.000 **	0.010 *
Chlorophyll content	Chla	0.001 **	0.000 **	0.000 **
	Chlb	0.009 **	0.000 **	0.004 **
	Chl	0.001 **	0.000 **	0.000 **
Gas exchange parameters	Pn	0.000 **	0.000 **	0.011 *
	Gs	0.035 *	0.000 **	0.594
	Ci	0.937	0.000 **	0.010 *
	Tr	0.000 **	0.000 **	0.055
Antioxidant enzyme	POD	0.976	0.000 **	0.139
	SOD	0.009 **	0.011 *	0.356
	CAT	0.000 **	0.000 **	0.000 **
	MDA	0.000 **	0.000 **	0.015 *

Note: * and ** indicate significant difference (*p* < 0.05) or extremely significant difference (*p* < 0.01). PH, plant height; RL, root length; RFW, root fresh weight; SD, stem diameter; LA, leaf area; Chla, chlorophyll a; Chlb, chlorophyll b; Chl, total chlorophyll; Pn, net photosynthetic rate; Gs, stomatal conductance; Ci, intercellular CO_2_ concentration; Tr, transpiration rate; POD, peroxidase; SOD, superoxide dismutase; CAT, catalase; MDA, malondialdehyde.

## Data Availability

The original contributions presented in this study are included in the article. Further inquiries can be directed to the corresponding author.

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
