# Peer review of "Effects of Polyethylene and Polystyrene Microplastics on Oat (*Avena sativa* L.) Growth and Physiological Characteristics"

_plants, 2025, doi:10.3390/plants15010056_

Round 1

Reviewer 1 Report

Comments and Suggestions for Authors
General Summary

The manuscript investigates the impacts of two common microplastic (MP) polymers—polyethylene (PE) and polystyrene (PS), both with a diameter of 2 μm—on the growth and physiological parameters of oat (Avena sativa L.). The authors aim to address a knowledge gap concerning the toxicity of MPs on feed crops by comparing different concentrations and polymer types. The study hypothesizes that MPs exert toxic effects that vary depending on type and concentration, and the results suggest that seedling growth is inhibited via stress responses.

Major Issues

1. The study evaluates only antioxidant enzymes (e.g., SOD, CAT) and chlorophyll content. This approach is insufficient for a comprehensive ecotoxicological assessment following microplastic exposure. It is recommended to include additional indicators, such as changes in plant secondary metabolites (e.g., flavonoids). Flavonoids can mitigate excessive ROS accumulation and oxidative stress induced by microplastics; thus, such indicators would more comprehensively reflect physiological and biochemical changes.

2. Microplastic concentrations were set at 0%, 0.1%, 0.5%, 1%, and 5% (w/w). It is essential for the authors to justify whether these concentrations (e.g., 0.1% and above) are environmentally relevant or if they represent a “worst-case” scenario. The authors should clearly state the rationale for choosing these concentrations in the Materials and Methods section, and in the Discussion section, explicitly compare these levels to the actual soil abundances reported in the Introduction (see references 30, 31, 32).

3. The conclusion currently states: "However, the primary mechanism underlying microplastic-induced toxicity remains unclear and warrants further investigation." For a scientific journal article (e.g., in Plants), merely reporting that toxicity occurred is insufficient; the authors must explain, based on their data, how toxicity occurred. For example: Did oxidative stress (as indicated by antioxidant enzyme activity) correlate with reduced biomass? Was there any evidence of physical blockage of root pores? Did MPs alter soil nutrient availability (e.g., N/P content)? The Discussion section should be significantly strengthened—not just describing results, but also linking physiological changes (e.g., chlorophyll content, enzyme activity) to observed growth phenotypes. If the mechanism remains “unclear,” the authors should at least propose a hypothesis based on their specific data (e.g., “The reduction in chlorophyll suggests that PS may interfere with…”), rather than leaving the issue unresolved.

4. The hypothesis states that toxicity varies with MP type. The manuscript should clearly explain why one polymer might be more toxic than the other. Is this due to differences in surface charge, particle shape, or the leaching of additives? The authors should discuss the physicochemical differences between PE and PS (e.g., aromatic rings in PS vs. aliphatic chains in PE) and how these differences might impact interactions with plant roots or soil aggregates.

Specific Comments

Abstract:

The abstract clearly identifies the gap (few studies on feed crops), but it should also briefly summarize key quantitative results (e.g., “Biomass decreased by X% under PS treatment”) rather than only describing the effects qualitatively.

Introduction:

Please ensure a clear transition between the discussion of “soil physicochemical properties” and “direct plant growth” effects. Microplastics can influence plants both indirectly (via changes to soil properties) and directly (via physical blockage or uptake). The authors should clarify which pathway(s) are the focus of this study.

Methodology (Microplastic Sizing):

The authors use 2 μm particles. Please clarify whether these are primary microplastics (commercially produced spheres) or secondary MPs (derived from ground or fragmented plastics), as this affects particle surface area and toxicity.

Terminology:

Replace all instances of “CK” or “Ck” with “Ctrl” or “Control” throughout the manuscript, including in figures and tables, for clarity and consistency.

References:

Reference formatting appears consistent, but please double-check for proper capitalization of article titles (e.g., see references 17, 33, and 64).

Reviewer 2 Report

Comments and Suggestions for Authors

This study investigated how two common soil microplastics—polyethylene (PE) and polystyrene (PS), both 2 µm in size—affect the growth, photosynthesis, and antioxidant concentration of oat seedlings (Avena sativa L.) when added to soil at different concentrations (0.1%, 0.5%, 1%, 5% w/w). It concluded that microplastics—especially polyethylene (PE) at high concentrations—significantly impair oat growth by inhibiting photosynthesis and disrupting antioxidant defenses, leading to oxidative damage. PS has milder and more variable effects, even stimulating photosynthesis at very low levels. The study highlights the need for further research on microplastic toxicity mechanisms and implications for agricultural soils and feed crop production.

There are some issues that must be addressed and corrected:

Keywords. Repeat words already in Title. "photosynthetic indices" appears only twice, in keywords and in table 1. There is no explanation for it and in fact it does not have any meaning.

Captions for the figures are incomplete.

"antioxidant system". There is no mention of "system" but rather concentrations of antoxidants.

The Discussion frequently restates the numerical or descriptive findings already presented in the Results. Examples: Re-describing how PE inhibited plant height, root length, etc., Re-explaining that PS at 0.1% increased photosynthesis.

The Discussion cites many studies and explains what others found in other species, while giving relatively little mechanistic interpretation based specifically on this oat dataset. Examples:  Comparing effects in tomatoes, cucumbers, wheat, fir trees, Citing ROS mechanisms from other studies. Uses literature to explain phenomena but does not analyze whether this experiment’s results support, contradict, or refine those mechanisms.

The manuscript does not fully integrate physiological datasets of Growth, Photosynthesis and Antioxidant "system" (concentration).

Missing are words of caution due to pot experiment limitations and short duration of experiment.

Also missing are the implications for agricultural practice, feed safety concerns, ecological  and policy interventions.

Comments on the Quality of English Language

The manuscript contains numerous English issues that will not be detailed here. Just general comments.
Frequent grammatical errors include: Incorrect singular/plural forms
Articles missing (“the”, “a”, “an”)
Subject–verb agreement errors

Many sentences repeat the same words or ideas unnecessarily., such as:
Significantly inhibited… significantly decreased… significantly reduced…” is repeated excessively.
Repeating “microplastics” multiple times in the same sentence.

Inconsistent tense. Use consistently primarily past tense for results.

Some terms are not suitable for scientific English. “potent inhibition” should be “strong inhibition”

Reviewer 3 Report

Comments and Suggestions for Authors

Please read the file

Round 2

Reviewer 1 Report

Comments and Suggestions for Authors

I am not entirely satisfied with the response to Comment 1. Given that the study concludes microplastics induce oxidative damage, it is crucial to recognize that plant defense mechanisms against such stress are multifaceted. They involve not only the antioxidant enzymes assessed here but also non-enzymatic systems, such as secondary metabolites (e.g., flavonoids). While I acknowledge the authors' plan to address this in future research, the current manuscript remains logically incomplete without acknowledging these alternative pathways. Therefore, I strongly suggest that the authors expand the Discussion section. Even in the absence of new experimental data, the authors must incorporate a literature-based discussion on how non-enzymatic antioxidants (like flavonoids) contribute to mitigating microplastic-induced oxidative stress. This addition is essential to provide a holistic view of the plant's stress response and to ensure the scientific soundness of the paper.

Reviewer 2 Report

Comments and Suggestions for Authors

This study investigated how two common soil microplastics affect the growth, photosynthesis, and antioxidant systems of oat seedlings (Avena sativa L.) when added to soil at different concentrations. IT concludes that microplastics—especially polyethylene (PE) at high concentrations—significantly impair oat growth by inhibiting photosynthesis and disrupting antioxidant defenses, leading to oxidative damage. The study highlights the need for further research on microplastic toxicity mechanisms and implications for agricultural soils and feed crop production.

Keywords repeat words already in Title. Photosynthetic indices appear in Keywords and in Table 1; it is not in M&M and its meaning is not clear.

Captions for Figures 3 to 5 are incomplete.

Uniform-sized oats were selected (Line 496) is incorrect. It should be seeds were selected.

Antioxidant system (Line 237) is not subject of this study. Antioxidant enzyme activity is the subject.

The manuscript is clear, objectives are well defined, methodologies are aligned with the objectives.

Comments on the Quality of English Language

The manuscript contains numerous English issues that I will not detail, just general observations.
Frequent grammatical errors include: Incorrect singular/plural forms, articles missing (“the”, “a”, “an”), subject–verb agreement errors

Many sentences repeat the same words or ideas unnecessarily. Significantly inhibited… significantly decreased… significantly reduced…” is repeated excessively.
Repeating “microplastics” multiple times in the same sentence.

Inconsistent tense. Use consistently primarily past tense for results.

Some terms are not suitable for scientific English. “potent inhibition” should be “strong inhibition”.

Reviewer 3 Report

Comments and Suggestions for Authors

Thank you for the careful revision of the manuscript. I have reviewed all the changes, and they address my previous comments satisfactorily. 

Author Response

Thank you for reviewing the manuscript

Round 3

Reviewer 1 Report

Comments and Suggestions for Authors

I have no further questions.